REGISTERED REPORT PROTOCOL

# Effectiveness of the pelvic floor muscle training on muscular dysfunction and pregnancy specific urinary incontinence in pregnant women with gestational diabetes mellitus: A systematic review protocol

**Angélica Mércia Pascon Barbosa**[1☯‡*], **Eusebio Mario Amador Enriquez**[2☯‡], **Meline Rossetto Kron Rodrigues**[3☯], **Caroline Baldini Prudencio**[2☯], **Álvaro Nagib Atallah**[4☯], **David Rafael Abreu Reyes**[2☯], **Raghavendra Lakshmana Shetty Hallur**[2☯], **Sthefanie Kenickel Nunes**[2☯], **Fabiane Affonso Pinheiro**[2☯], **Carlos Isaías Sartorão Filho**[2☯], **Gabriela Lopes Piemonte Andrade**[5☯], **Bary Berghmans**[6☯], **Rob de Bie**[6☯], **Silvana Andréa Molina Lima**[7☯‡], **Marilza Vieira Cunha Rudge**[2☯‡], **The Diamater Study Group**[¶]

1 Department of Physiotherapy and Occupational Therapy, School of Philosophy and Sciences, São Paulo State University (UNESP), Marilia, São Paulo State, Brazil, 2 Department of Gynecology and Obstetrics, Botucatu Medical School (FMB), São Paulo State University (UNESP), Botucatu, São Paulo State, Brazil, 3 Stricto Sensu Graduate Program in Nursing at UNIVERITAS, Guarulhos University (UNG), Guarulhos, São Paulo, Brazil, 4 Discipline of Evidence-Based Medicine, Paulista School of Medicine-Federal University of São Paulo (EPM-UNIFESP), São Paulo, Brazil, 5 Department of Physiotherapy, Universidade do Oeste Paulista (UNOESTE), Presidente Prudente, São Paulo State, Brazil, 6 Pelvic Care Center Maastricht, Maastricht University Medical Center, Maastricht, The Netherlands, 7 Department of Nursing, Botucatu Medical School (FMB), São Paulo State University (UNESP), Botucatu, São Paulo State, Brazil

☯ These authors contributed equally to this work.
‡ AMPB and EMAE are first authors on this work. SAML and MVCR are last authors on this work.
¶ Membership of The Diamater Study Group is listed in the Acknowledgments.
* angelicapascon@gmail.com

## Abstract

### Background

There is ample evidence that gestational diabetes mellitus has a direct influence on urinary incontinence and pelvic floor muscles. There are no standardized pelvic floor muscle exercise programs in the literature for the physiotherapy and differ in the type of exercise, intensity, type and duration of application, and the frequency and duration of treatment sessions. The aim of this systematic review will be to investigate that Pelvic Floor Muscle Training can prevent and/or decrease the pregnancy specific urinary incontinence in women with gestational diabetes mellitus or gestational hyperglycemia.

### Methods

We will perform a systematic review according to the Cochrane methodology of Randomized Controlled Trials. An overall search strategy will be developed and adapted for Embase, MEDLINE, LILACS, and CENTRAL databases, with the date of consultation until

**Data Availability Statement:** All relevant data from this study will be made available upon study completion.

**Funding:** This ongoing (2018-2020) project (FAPESP 2018/17534-1) is funded by the São Paulo Research Foundation-FAPESP, government agency located in São Paulo, Brazil. Linked with the thematic project Fundação de Amparo à Pesquisa do Estado de São Paulo, FAPESP 2016/01743-5. The funders had not and will not have a role in study design, data collection and analysis, decision to publish, or preparation of the manuscript.

**Competing interests:** The authors have declared that no competing interests exist.

**Abbreviations:** UI, Urinary incontinence; SUI, Stress urinary incontinence; GDM, Gestational diabetes mellitus; PFM, Pelvic floor muscles; PFMD, Pelvic floor muscle dysfunction; RCTs, Randomized controlled trials; CI, Confidence interval; PS-UI, Pregnancy specific urinary incontinence.

June 2020. The MeSH terms used will be "Pregnancy", "Hyperglycemia", "Diabetes Mellitus, Type 2", "Diabetes Mellitus, Type 1", "Pregnancy in Diabetics", "Diabetes, Gestational", "Urinary Incontinence", "Pelvic Floor Muscle Strength". *Primary outcomes*: improvement or cure of pregnancy specific urinary incontinence (which can be assessed by questionnaires, and tools such as tampon test, voiding diary, urodynamic study). *Secondary outcomes*: improvement of pelvic floor muscle strength (pelvic floor functional assessment, perineometer, electromyography, functional ultrasonography), improved quality of life (questionnaires), presence or absence of postpartum Urinary Incontinence and adverse effects. Quality assessment by Cochrane instrument. Metanalysis if plausible, will be performed by the software Review Manager 5.3.

## Discussion

The present study will be the first to analyze the effectiveness of pelvic floor exercises in pregnant women with Gestational Diabetes Mellitus or Hyperglycemia, who suffer from pregnancy specific urinary incontinence. Randomized Controlled Trials design will be chosen because they present the highest level of evidence. It is expected to obtain robust and conclusive evidence to support clinical practice, in addition to promoting studies on the theme and contributing to new studies.

## Trial registration

**Systematic review registration:** PROSPERO CRD42017065281.

## Background

Each day it is more obvious that Urinary Incontinence (UI) in pregnant women is associated, among other factors, with Gestational Diabetes Mellitus (GDM) and Hyperglycemia in Pregnancy. Clinical findings of high prevalence of UI in women with GDM, two years before labor and prior cesarean section, provide evidence of the need to increase knowledge of this association: Prior GDM and UI [1]. The association between GDM and the increased prevalence of UI and pelvic floor muscle dysfunction (PFMD) was clearly demonstrated by Barbosa *et al*. (2011) who concluded that the prevalence of pregnancy specific urinary incontinence (PS-UI) and UI two years postpartum was significantly higher among women with GDM than among normoglycemic pregnant women. Their multivariate analysis demonstrated that GDM is an independent risk factor for the occurrence of PS-UI and they showed that pregnant women with prior GDM have weakness of the pelvic floor muscles (PFM) two years after birth [2].

Santos *et al*. (2006) stated that stress urinary incontinence (SUI) is the most common and prevalent type of UI during the reproductive years [3]. This type of UI is defined as involuntary loss of urine during physical exertion, due to a sudden increase in intra-abdominal pressure in the absence of detrusor contraction or underactivity [4]. The etiology of SUI is not fully understood, but injuries to the pelvic floor during pregnancy and childbirth are suggested as main risk factors [3]. Although the SUI is the most prevalent UI type during pregnancy, women also complain about urgency urinary incontinence and its combination named mixed urinary incontinence [5].

For many years involuntary loss of urine in pregnant women have complaint more often to her doctor or physiotherapist. Although this increase in cases reported by patients was not as

significant as it is today, it was enough for many researchers worldwide to be interested to assess, evaluate, report and look for solutions to prevent or reduce this discomfort in women [6]. A Cochrane review from 2018 considered PFM training as first-line conservative management to SUI and all other types of UI nevertheless they didn't include PS-UI treatment on the revision [7]. Another Cochrane review from 2020 showed that due to the quality and quantity of well designed studies when antenatal or postnatal populations were evaluated the influence of PFM training for treatment and/or prevetion is uncertain or not at all clear [5]. So a little is know about UI during pregnancy, particularly the impact of the diabetes status on the PS-UI [5].

Considering this background, the aim of our study is, using a systematic review of available randomized clinical trials (RCTs), to investigate in GDM pregnant women with gestational hyperglycemia the effect of PFM training for the prevention and/or treatment of PS-UI.

## Methods/design

### Systematic review

**Study inclusion and exclusion criteria.** *Inclusion criteria.* RCTs of pregnant women with hyperglycemia (all levels), any type of UI developed during pregnancy, nulliparous, primiparous and multiparous women during pregnancy, receiving PFM training to prevent and/or reduce the symptoms of PS-UI.

*Exclusion criteria.* All other designs, not being RCT.

**Outcomes.** *Primary outcomes.* Occurrence, reduction, recurrence, or persistence of PS-UI symptoms.

*Secondary outcomes.* Improvement of PFM strength (pelvic floor functional assessment, perineometer, electromyography, functional ultrasonography), improved quality of life (questionnaires), presence or absence of postpartum UI and adverse effects.

**Literature search strategies.** A model strategy will be created and adapted for searching the data bases Embase, Pubmed, Lilacs and CENTRAL to identify studies involving the abovementioned interventions. RCTs registry sites will be consulted for ongoing studies on the topic (Rebec, Clinicaltrial.gov). Grey literature will be consulted as well as Capes database. Last date of searche will be 30 June 2020. There will be no language restrictions. MeSH terms used for our searches are "Pregnancy", "Hyperglycemia", "Diabetes Mellitus, Type 2", "Diabetes Mellitus, Type 1", "Pregnancy in Diabetics", "Diabetes, Gestational", "Urinary Incontinence" and "Pelvic Floor Muscle Strength".

**Data collection and analysis.** *Selection of studies.* For each search strategy, two reviewers will independently evaluate the studies retrieved from the databases in the order: title, abstract and full reading. All studies potentially eligible for inclusion in the review will be selected for full reading. In case of disagreement, a third reviewer will be consulted.

*Data extraction and management.* Data extraction for eligible studies will be performed by two reviewers (EMAE e MRKR) who will independently extract data from articles that meet the inclusion criteria. A standardized form will be used to extract the following information: study characteristics (design, randomization method, blinding, allocation generation and concealment, statistics); participants; interventions (definition of exercises); clinical outcomes (types of outcomes measured: dichotomous or continuous and adverse effects) (Tables 1 and 2).

**Table 1. Study characteristics related to the number of participants, inclusion, and exclusion criteria.**

| Author/ Year | No. of participants | Inclusion criteria | Exclusion criteria |
|---|---|---|---|
|  |  |  |  |
|  |  |  |  |

**Table 2. Study characteristics related to the configuration; the number of participants according to the group; gestational age; exercise definition, outcome measures for women; and follow up.**

| Title, Authors, Years | Location | Number of participants per group | Gestational age (weeks) (baseline) | Definition of exercises (training protocol details) | Outcomes | Follow-up (simplified) | Results | Grade Score |
|---|---|---|---|---|---|---|---|---|
|  |  |  |  |  |  |  |  |  |
|  |  |  |  |  |  |  |  |  |

Studies that do not fully meet the inclusion criteria will be excluded and tabulated with their justification for the exclusion (Table 3).

**Methodological quality, risk of bias and statistical report.** The risk of bias will be assessed according to the Cochrane Handbook of Interventions Systematic Reviews, which assesses the following domains: allocation generation; concealment of allocation; masking (of participants and researchers); the presence of incomplete data; reporting bias of information and other types of bias. The answers to these domains may be "Yes", "No" or "Uncertain". The final grade of the study will be based on the responses given to the first three domains and will be classified as having high, low or risk of uncertain bias [8]. Dichotomous data will be calculated relative risk, i.e. proportion of events in the treatment group under the relation to the proportion of events in the control group, with a 95% Confidence Interval (CI). Such estimates will be calculated from the intention-to-treat analysis approach. Continuous data will be expressed as means and standard deviation, and the weighted average proportion with 95% CI will be calculated. Publication bias will be evaluated with funnel plots and formally assessed with the Egger test. For variability in results between studies, the I2 statistic and the P-value obtained from the Chi-squared Cochrane test will be used. Review Manager software (RevMan) will be used for all analyses, including meta-analysis if possible [9–11].

**Quality of evidence.** We intend to use the Evaluation, Development and Evaluation Recommendation Rating (GRADE) to assess the overall strength of evidence assessment. In RCT, the GRADE system evaluates the limitations of the study, inconsistencies, indirect evidence, inaccuracies, and publication bias, classifying the evidence as high, moderate, low, or very low [12].

## Discussion

The literature suggest that PFM exercises, accompanied by training of approximately 12 weeks including aerobic and resistance exercises under intense supervised guidance, are effective for prevention, treatment and reduction of PS-UI in women with GDM and/or HG [13–15]. However, there is no systematic review that evaluated the standardized PFM Exercises program for physiotherapy.

In literature there are RCTs, but not summarized. This study design has been chosen because in this way we may assess the highest available level of evidence. The present systematic review will be the first to analyze the effectiveness of PFM exercises in pregnant women with GDM or HD with PSUI.

**Table 3. Exclusion table from systematic review and meta-analysis of randomized controlled trials to evaluate the effects of PFM exercise on pregnant women with GDM and gestational urinary incontinence and/or pelvic floor muscle dysfunction.**

| First author, Years of publication | Title | Reasons for exclusion |
|---|---|---|
|  |  |  |
|  |  |  |

In this way we may obtain robust and conclusive evidence whether or not there is evidence to support clinical practice, in addition to promote high quality studies on the subject.

## Supporting information

**S1 File. PRISMA-P (Preferred Reporting Items for Systematic review and Meta-Analysis Protocols) 2015 checklist: Recommended items to address in a systematic review proto-col**∗.
(DOC)

**S2 File. PROSPERO international prospective register of systematic reviews.**
(PDF)

## Acknowledgments

We thank the Diamater Study Group for making valuable comments.

**The Diamater Study Group:** Rudge MVC, Barbosa, AMP, Caldeiron IMP, Souza FP, Berghmans B, de Bie R, Thabane L, Junginger B, Graeff CFO, Magalhães CG, Costa RA, Lima SAM, Kron-Rodrigues MR, Felisbino S, Barbosa W, Campos FJ, Bossolan G, Corrente JE, Nunes HRC Abbade J, Rossignoli PS, Pedroni CR, Atallah AN, Di Bella ZIKJ, Uchoa SMM, Hungaro MA, Mareco EA, Sakalem ME, Martinho N, Hallur LSR, Reyes DRA, Alves FCB, Marcondes JPC, Prudencio CB, Pinheiro FA, Sartorão CI, Quiroz SBCV, Pascon T, Nunes SK, Catinelli BB, Reis FVDS, Oliveira RG, Barneze S, Enriquez EMA, Takano L, Carr AM, Magyori ABM, Iamundo LF, Carvalho CNF, Jacomin M, Avramidis RE, Silva AJB, Orlandi MIG, Dangió TD, Bassin HCM, Melo JVF, Takemoto MLS, Menezes MD, Caldeirão TD, Santos NJ, Lourenço IO, Marostica de Sá J, Caruso IP, Rasmussen LT, Garcia GA, Nava GTA, Pascon C, Bussaneli DG, Nogueira VKC, Rudge CVC, Piculo F, Prata GM.

## Author Contributions

**Conceptualization:** Angélica Mércia Pascon Barbosa, Eusebio Mario Amador Enriquez, Meline Rossetto Kron Rodrigues, Álvaro Nagib Atallah, Silvana Andréa Molina Lima, Marilza Vieira Cunha Rudge.

**Data curation:** Angélica Mércia Pascon Barbosa, Eusebio Mario Amador Enriquez, Meline Rossetto Kron Rodrigues, Silvana Andréa Molina Lima, Marilza Vieira Cunha Rudge.

**Formal analysis:** Angélica Mércia Pascon Barbosa, Eusebio Mario Amador Enriquez, Meline Rossetto Kron Rodrigues, Silvana Andréa Molina Lima, Marilza Vieira Cunha Rudge.

**Funding acquisition:** Angélica Mércia Pascon Barbosa, Marilza Vieira Cunha Rudge.

**Investigation:** Angélica Mércia Pascon Barbosa, Eusebio Mario Amador Enriquez, Meline Rossetto Kron Rodrigues, David Rafael Abreu Reyes, Raghavendra Lakshmana Shetty Hallur, Sthefanie Kenickel Nunes, Silvana Andréa Molina Lima, Marilza Vieira Cunha Rudge.

**Methodology:** Angélica Mércia Pascon Barbosa, Eusebio Mario Amador Enriquez, Meline Rossetto Kron Rodrigues, Silvana Andréa Molina Lima, Marilza Vieira Cunha Rudge.

**Project administration:** Angélica Mércia Pascon Barbosa, Meline Rossetto Kron Rodrigues, Silvana Andréa Molina Lima, Marilza Vieira Cunha Rudge.

**Resources:** Angélica Mércia Pascon Barbosa, Eusebio Mario Amador Enriquez, Marilza Vieira Cunha Rudge.

**Software:** Angélica Mércia Pascon Barbosa, Eusebio Mario Amador Enriquez, Meline Rossetto Kron Rodrigues, Silvana Andréa Molina Lima, Marilza Vieira Cunha Rudge.

**Supervision:** Angélica Mércia Pascon Barbosa, Marilza Vieira Cunha Rudge.

**Validation:** Angélica Mércia Pascon Barbosa, Marilza Vieira Cunha Rudge.

**Visualization:** Angélica Mércia Pascon Barbosa, Eusebio Mario Amador Enriquez, Meline Rossetto Kron Rodrigues, David Rafael Abreu Reyes, Raghavendra Lakshmana Shetty Hallur, Sthefanie Kenickel Nunes, Fabiane Affonso Pinheiro, Carlos Isaías Sartorão Filho, Gabriela Lopes Piemonte Andrade, Bary Berghmans, Rob de Bie, Silvana Andréa Molina Lima, Marilza Vieira Cunha Rudge.

**Writing – original draft:** Angélica Mércia Pascon Barbosa, Eusebio Mario Amador Enriquez, Meline Rossetto Kron Rodrigues, David Rafael Abreu Reyes, Raghavendra Lakshmana Shetty Hallur, Sthefanie Kenickel Nunes, Fabiane Affonso Pinheiro, Carlos Isaías Sartorão Filho, Gabriela Lopes Piemonte Andrade, Bary Berghmans, Rob de Bie, Silvana Andréa Molina Lima, Marilza Vieira Cunha Rudge.

**Writing – review & editing:** Angélica Mércia Pascon Barbosa, Eusebio Mario Amador Enriquez, Meline Rossetto Kron Rodrigues, Caroline Baldini Prudencio, Raghavendra Lakshmana Shetty Hallur, Fabiane Affonso Pinheiro, Carlos Isaías Sartorão Filho, Gabriela Lopes Piemonte Andrade, Bary Berghmans, Rob de Bie, Silvana Andréa Molina Lima, Marilza Vieira Cunha Rudge.

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
