## [Decision Letter · Decision Letter 0]

28 Sep 2020

PONE-D-20-10898

Effectiveness of the pelvic floor muscle training on muscular dysfunction and pregnancy specific urinary incontinence in pregnant women with gestational diabetes mellitus: A systematic review protocol

PLOS ONE

Dear Dr. Barbosa,

Thank you for submitting your manuscript to PLOS ONE. After careful consideration, we feel that it has merit but does not fully meet PLOS ONE’s publication criteria as it currently stands. Therefore, we invite you to submit a revised version of the manuscript that addresses the points raised during the review process.

SPECIFIC ACADEMIC EDITOR COMMENTS: An expert in the field handled your manuscript, and we are very thankful for their time and efforts. Although interest was found in your study, there were some concerns that arose needing to be addressed by the authors. Notably, it would benefit the manuscript to expand on the type of PFM training and duration and how this affects the studied outcomes; there are comments about the MESH terms and keyword search strategy; and there are questions about the experiment design, including specifics about the exclusion and inclusion criteria, quality score and quality check of the evidence, and subgroup comparisons. Please address ALL of the reviewer's comments in your revised manuscript.

We look forward to receiving your revised manuscript.

Kind regards,

Frank T. Spradley

Academic Editor

PLOS ONE

2. During our internal pre-review checks, we noticed some errors of English grammar and use in this manuscript. However, we felt that the language quality is sufficient to allow a scientific assessment of the manuscript. If, based on your evaluation and the reviewer’s comments, you feel a revision or accept decision is merited, we will ask authors to copyedit the manuscript at the time of first decision. If you have any questions or concerns, or if language concerns persist following revision, please email Associate Editor Nancy Beam at nbeam@plos.org.

3.We note that you have stated that you will provide repository information for your data at acceptance. Should your manuscript be accepted for publication, we will hold it until you provide the relevant accession numbers or DOIs necessary to access your data. If you wish to make changes to your Data Availability statement, please describe these changes in your cover letter and we will update your Data Availability statement to reflect the information you provide.

4.Thank you for stating the following in the Funding Section of your manuscript:

[The funding for this systematic review protocol (part of a Masters degree project) is provided

by the São Paulo Research Foundation (FAPESP), São Paulo, Brazil and its grant number is

FAPESP 2018/17534-1. FAPESP is a government funding agency with the aim of providing

grants, funds, and programs to support research, education, and innovation. This study is

linked to the thematic project “The new gestational triad: hyperglycemia, urinary incontinence (UI)

and biomolecular profile as a long-term predictor for UI; a prospective cohort study in translational

research with biodevice with stem cells for muscle regeneration in diabetic rats” and its linked

Process number is FAPESP 2016/01743-5.]

 [The funders had and will not have a role in study design, data collection and analysis, decision to publish, or preparation of the manuscript.]

5.One of the noted authors is a group or consortium [The Diamater Study Group]. In addition to naming the author group, please list the individual authors and affiliations within this group in the acknowledgments section of your manuscript. Please also indicate clearly a lead author for this group along with a contact email address.

Reviewers' comments:

Reviewer's Responses to Questions

**Comments to the Author**

1. Does the manuscript provide a valid rationale for the proposed study, with clearly identified and justified research questions?

Reviewer #1: Yes

2. Is the protocol technically sound and planned in a manner that will lead to a meaningful outcome and allow testing the stated hypotheses?

Reviewer #1: Yes

3. Is the methodology feasible and described in sufficient detail to allow the work to be replicable?

Reviewer #1: Yes

4. Have the authors described where all data underlying the findings will be made available when the study is complete?

Reviewer #1: Yes

5. Is the manuscript presented in an intelligible fashion and written in standard English?

Reviewer #1: Yes

6. Review Comments to the Author

You may also provide optional suggestions and comments to authors that they might find helpful in planning their study.

Reviewer #1: Very good paper. It addresses an issue that is neglected in many societies.

I have a few minor points. Hope these comments will be helpful to the authors.

Please read the following information and revise your manuscript as necessary.

Please explain more about the type of PFM training and the duration of it in detail.

The MeSH terms used will be "Pregnancy", "Hyperglycemia", "Diabetes Mellitus, Type 2", "Diabetes Mellitus, Type 1", "Pregnancy in Diabetics", "Diabetes, Gestational" and "Urinary Incontinence".

The search strategy does not mention the keywords "pelvic floor muscle strength" and "quality of life". I have the impression that it is better to mention them.

The quality score of the articles is not recorded in the item tables of study characteristics. It is better to added.

Who are two independent investigators to check the quality of evidence?

It not clear if all types of urinary incontinence in pregnancy considered or stress urinary incontinence? It is not explicitly mentioned in study inclusion and exclusion criteria.

If yes. Do the authors intend to compare subgroups?

Given that the type of PFM training and the duration of it affects the outcome. How can this be considered? It is better to be explained.

With respect

7. PLOS authors have the option to publish the peer review history of their article (what does this mean?). If published, this will include your full peer review and any attached files.

Reviewer #1: No

---

## [Author Response · Author response to Decision Letter 0]

1 Oct 2020

The response bellow is also attached in a file named "response to reviewers"

2020.30.09

Reply to Reviewer comment

Dear Editor

Thank you for giving me the opportunity to review this article.

Very good paper. It addresses an issue that is neglected in many societies.

I have a few minor points. Hope these comments will be helpful to the authors.

Please read the following information and revise your manuscript as necessary.

Dear reviewer, we really appreciate yours suggestions and we carefully reviewed and considered all topics to enhance our manuscript. Our answers on this response are in red. We added two co-authors to the main list due to their contribution to the conceptualization and the present revision. 

1) Please explain more about the type of PFM training and the duration of it in detail.

We really appreciate your comment and to provide further rationality about PFM training we added informations in third paragraph on introduction. Additionally, we specified in the table 2 the column that the training protocol details will be fill.

2) The MeSH terms used will be "Pregnancy", "Hyperglycemia", "Diabetes Mellitus, Type 2", "Diabetes Mellitus, Type 1", "Pregnancy in Diabetics", "Diabetes, Gestational" and "Urinary Incontinence". The search strategy does not mention the keywords "pelvic floor muscle strength" and "quality of life". I have the impression that it is better to mention them.

Thank you for such interesting suggestion, after a careful group discussion we decided to include the keyword “pelvic floor muscle strength”, because we considered that it could be useful to improve our search strategy. 

However we decided not to include the keyword “quality of life”, we agreed that adding it to our strategy search, articles which could be interesting for our final aim but didn’t measure the impact on this field could be excluded by this choice.

3) The quality score of the articles is not recorded in the item tables of study characteristics. It is better to added.

We appreciate this recommendation and we added on table 2 a column with grade score.

4) Who are two independent investigators to check the quality of evidence?

Thank you, we added the name initials on the manuscript (EMAE e MRKR).

5) It not clear if all types of urinary incontinence in pregnancy considered or stress urinary incontinence? It is not explicitly mentioned in study inclusion and exclusion criteria. If yes. Do the authors intend to compare subgroups?

Thank you, we added a paragraph in the introduction session addressing other types of IU than SUI. We don’t have intention to compare subgroups and because of it we found better to include “any type of UI developed during pregnancy” in the inclusion criteria.

6) Given that the type of PFM training and the duration of it affects the outcome. How can this be considered? It is better to be explained.

Thank you, we added definition of exercises (training protocol details) on table 2 and on the method session line 127. On this column should be full-filled informations about repetition, duration, intensity (and other important information). All those characteristics of the training protocol, will be considerer to the final analyses.

---

## [Editor Report · Decision Letter 1]

26 Oct 2020

Effectiveness of the pelvic floor muscle training on muscular dysfunction and pregnancy specific urinary incontinence in pregnant women with gestational diabetes mellitus: A systematic review protocol

PONE-D-20-10898R1

Dear Dr. Barbosa,

We’re pleased to inform you that your manuscript has been judged scientifically suitable for publication and will be formally accepted for publication once it meets all outstanding technical requirements.

Kind regards,

Frank T. Spradley

Academic Editor

PLOS ONE

---

## [Editor Report · Acceptance letter]

13 Nov 2020

PONE-D-20-10898R1 

Effectiveness of the pelvic floor muscle training on muscular dysfunction and pregnancy specific urinary incontinence in pregnant women with gestational diabetes mellitus: A systematic review protocol 

Dear Dr. Barbosa:

I'm pleased to inform you that your manuscript has been deemed suitable for publication in PLOS ONE. Congratulations! Your manuscript is now with our production department. 

Kind regards, 

on behalf of

Dr. Frank T. Spradley 

Academic Editor

PLOS ONE